# Demonstration of quantum-digital payments

Peter Schiansky [1,4], Julia Kalb[1,4], Esther Sztatecsny[1],
Marie-Christine Roehsner [1,3], Tobias Guggemos[1], Alessandro Trenti[1,3],
Mathieu Bozzio [1] ✉ & Philip Walther [1,2] ✉

Digital payments have replaced physical banknotes in many aspects of our daily lives. Similarly to banknotes, they should be easy to use, unique, tamper-resistant and untraceable, but additionally withstand digital attackers and data breaches. Current technology substitutes customers' sensitive data by randomized tokens, and secures the payment's uniqueness with a cryptographic function, called a cryptogram. However, computationally powerful attacks violate the security of these functions. Quantum technology comes with the potential to protect even against infinite computational power. Here, we show how quantum light can secure daily digital payments by generating inherently unforgeable quantum cryptograms. We implement the scheme over an urban optical fiber link, and show its robustness to noise and loss-dependent attacks. Unlike previously proposed protocols, our solution does not depend on long-term quantum storage or trusted agents and authenticated channels. It is practical with near-term technology and may herald an era of quantum-enabled security.

The development of quantum algorithms compromising modern cryptography has triggered global research for stronger security levels[1–3]: the security of current cryptographic schemes relies on computationally hard mathematical problems (known as computational security), which should be replaced by quantum-resistant schemes. While research and standardization for such quantum-resistant solutions are blossoming, some of them have already been broken by computational attacks[4–6].

Quantum-mechanical laws, on the other hand, can provide security against adversaries with unlimited computational power for some tasks[7,8]. This type of security, known as information-theoretic security (i.t.-security), is one of the motivations towards a quantum internet[9]. So far, Quantum Key Distribution (QKD) is the most mature and widely implemented quantum technology: it allows two mutually trusted parties to communicate securely over a public channel. QKD can already establish i.t.-secure connections over 500 km of optical fiber[10,11] and 1000 km of free space using satellites[12,13].

In the modern era of digital payments ranging from contactless purchases to online banking, a plethora of new security threats arise.

One significant threat occurs when customers interact with untrusted merchants, who may not have sufficient means to protect against external hackers, or may be malicious themselves[14]. In that case, a binding commitment between the customer, the merchant, and the bank or payment-network is required to guarantee the validity of a transaction. Such a bond usually comes in the form of a cryptogram[15,16], which is the output of a hash function that guarantees the one-time nature of each purchase. Since not all parties involved are trusted, QKD is not suitable to provide i.t.-security here, and other quantum solutions need to be established. Device-independent versions of QKD[17–19], which do not assume trusted quantum sources or detectors, are also inadequate, since the final classical output (i.e., the cryptogram) is handled by the untrusted parties themselves.

Motivated by the no-cloning property of quantum mechanics, previous works have investigated the potentials and drawbacks of using quantum light in the prevention of banknote counterfeiting[20–22] and double-spending with tokens or credit cards[23–27]. Introducing this fundamentally new type of money to everyday scenarios is, however, technologically challenging: quantum states must be stored over days

[1]University of Vienna, Faculty of Physics, Vienna Center for Quantum Science and Technology (VCQ), 1090 Vienna, Austria. [2]Christian Doppler Laboratory for Photonic Quantum Computer, Faculty of Physics, University of Vienna, 1090 Vienna, Austria. [3]Present address: Security and Communication Technologies, Center for Digital Safety and Security, AIT Austrian Institute of Technology GmbH, Giefinggasse 4, 1210 Vienna, Austria. [4]These authors contributed equally: Peter Schiansky, Julia Kalb. ✉e-mail: mathieu.bozzio@univie.ac.at; philip.walther@univie.ac.at

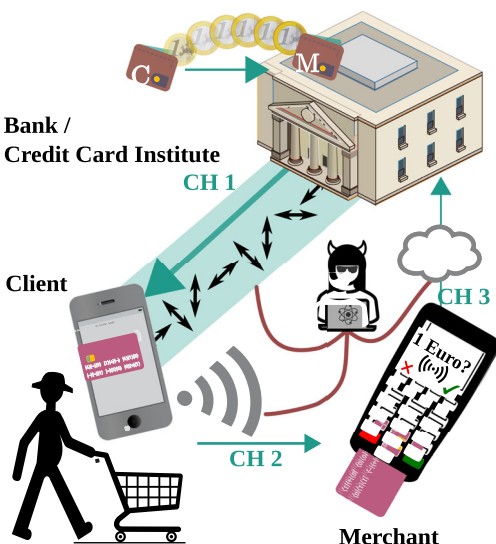

**Fig. 1 | Simplified representation of quantum-digital payments.** As in classical payments, we consider three parties: a Client, a Merchant, and a Bank/Creditcard institute. In contrast to ref. 32, we do not assume any quantum or classical communication channel to be trusted (i.e., CH 1, CH 2, and CH 3 are insecure), except an initial prior step between the Bank and Client for an account creation. All parties involved apart from the Bank can also act maliciously. During a payment, the Bank sends a set of quantum states to the Client's device (e.g., phone, computer, etc.), which measures them and transforms them into a quantum-secured payment token −cryptogram−which we display here as a one-time credit card. The Client uses this classical token for paying the Merchant, who then contacts the Bank for payment verification. If the payment is accepted, the bank transfers the money from the Client's account to the Merchant's.

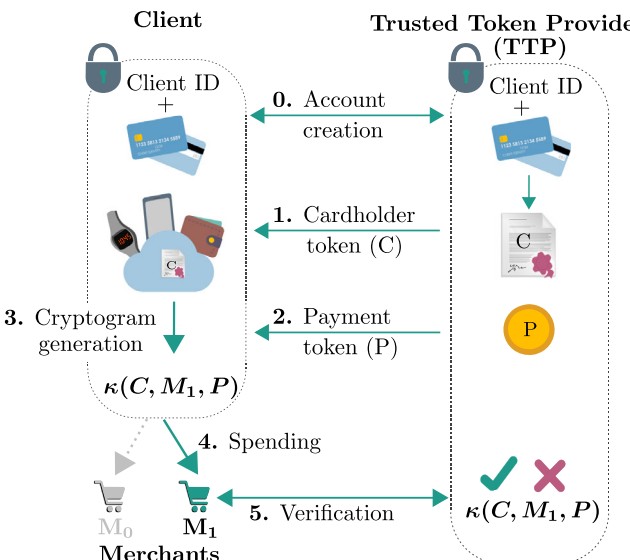

**Fig. 2 | Classical digital payments.** Step 0: The Client sets up an account at the Trusted Token Provider (TTP), providing their secret ID and sensitive credit card information through an authenticated and encrypted channel. Step 1: The Client authenticates with the TTP, and requests a cardholder token $C$, which the TTP sends through a secure channel. Step 2: The TTP randomly generates a one-time token $P$ and sends it to the Client through a secure channel. Step 3: The Client's device uses the stored secret token $C$, the public merchant ID $M_i$, and the payment token $P$ to compute a cryptogram $\kappa(C, M_i, P)$. Step 4: The Client spends the cryptogram at the chosen Merchant. Step 5: The Merchant verifies the cryptogram with the TTP, and accepts or rejects the transaction.

or months to ensure flexible spending. This is far beyond state-of-the-art quantum storage times, which range from a few microseconds to a few minutes[28–30]. Recently, an interesting alternative was proposed, replacing quantum storage by a network of trusted agents and authenticated channels, positioned at precise spatial locations with respect to the spending points[31,32]. From a practical standpoint, this approach presents new drawbacks, as customers and online shoppers do not have the means to securely set up complex trust networks for everyday transactions. Furthermore, accurately monitoring the spatial and temporal coordinates of verifiers requires a trusted Global Positioning System (GPS), which opens the door to undesired spoofing-type attacks[33].

In this work, we show how quantum light can provide practical security advantages over classical methods in everyday digital payments. As shown in Fig. 1, we generate and verify i.t.-secure quantum cryptograms, in such a way that the unforgeability and user privacy properties from previous experimental works holds[32], but all intermediate channels, networks, and parties are untrusted, thus significantly loosening the security assumptions. Only one authenticated communication (between the client and their payment provider) has to take place at an arbitrary prior point in time. The concealment of the customers' sensitive information is guaranteed by an i.t.-secure function, and the commitment to the purchase is guaranteed by the laws of quantum mechanics. Additionally, no cross-communication is required to validate the transaction in the case of multiple verifier branches. Our implementation is performed over a 641 m urban fiber link, and can withstand the full spectrum of noise and loss-dependent attacks, including those exploiting reporting strategies[34].

## Results
### Digital payments
We first describe the main security concepts of today's online and contactless purchases[15,16] (actual implementation may vary). Following

Fig. 2, each Client initially sets up an account with a Trusted Token Provider (TTP) via a secure communication channel. The TTP is usually the Client's bank, credit card provider, or a trusted external company. Through this initial step, the Client is assigned a unique identification token $C$, which is stored securely on both the Client's and TTP's devices. The Client's stored data can be, e.g., an electronic wallet or a virtual credit card stored on a smartphone, watch, etc.

When the Client wishes to purchase goods or services from a given Merchant $M_i$, it has to be ensured that malicious parties, including untrusted Merchants, cannot spend in the Client's name at another place or time. That is why the Client receives a one-time payment token $P$ from the Merchant or TTP, which is used to compute a *cryptogram*, an output of a function of their secret token $C$, the Merchant's public ID $M_i$, and the one-time payment token $P$. We note here that the Merchant ID $M_i$ must be valid and honest (e.g., provided by a Public Key Infrastructure or a securely pre-shared locally stored database). This cryptogram, which we call $\kappa(C, M_i, P)$, is communicated to the Merchant, who then sends it to the TTP for verification. The TTP can verify the signature and uniqueness of the cryptogram, since they have knowledge of all three inputs $C$, $M_i$, and $P$.

In real-world applications, the cryptogram is the output of a cryptographic hash- or encryption function[16,35] that is computationally secure. However, this would allow a malicious party with sufficient computational power to run through all input combinations of $C$, $P$, and $M_i$ until they recover the one combination that matches the original cryptogram. In that case, the Client's ID and payment data are completely compromised.

### Quantum advantage
Considering these attacks only, previous quantum-digital signature schemes can provide i.t.-security[36,37]. However, they typically require QKD channels and classical authentication between all three parties.

In this work, we propose a quantum solution that requires only one QKD for the initial step between Client and TTP (Step 1 in Fig. 2).

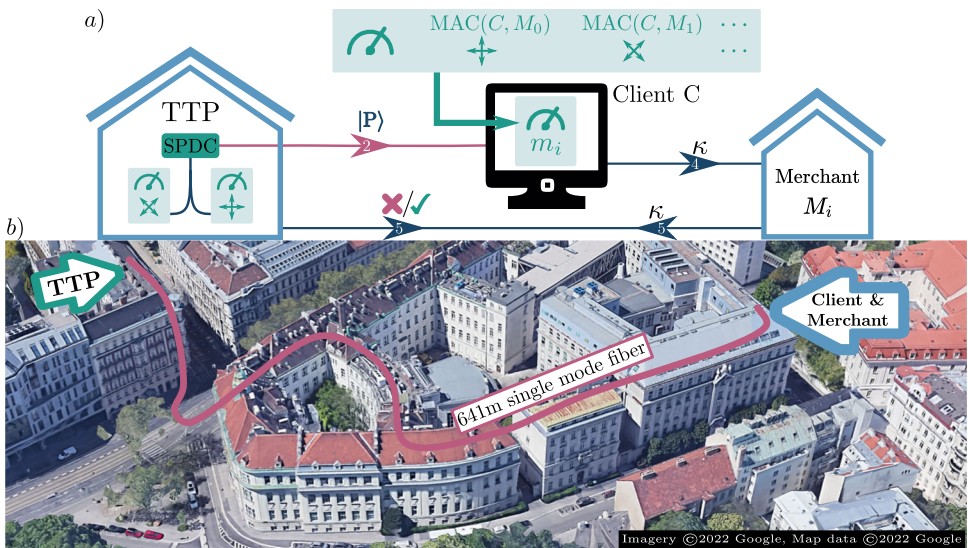

**Fig. 3 | Experimental quantum-digital payments. a** The Trusted Token Provider (TTP) creates entangled photon pairs using a Spontaneous Parametric Down Conversion (SPDC) source. One photon's polarization is randomly measured by the TTP in either a linear or diagonal basis, creating the classical description $(b, \mathcal{B})$, which remotely prepares the quantum token $|P\rangle$ on the second photon. The latter is sent to the Client through a 641 m long optical fiber link, who measures its polarization in a basis $m_i = MAC(C, M_i)$ specified by a Message Authentication Code

(MAC) of the Merchant's ID $M_i$ and the Client's private token $C$, and thereby obtains the *cryptogram* that is $\kappa_i \xleftarrow{m_i} |P\rangle$. Classical communication between the TTP, Client and Merchant is used to verify the compatibility of $\kappa$, $M_i$ and $C$ with $(b, \mathcal{B})$. Red (blue) lines indicate quantum (classical) channels. The arrow numbering indicates the steps from Fig. 2. **b** Satellite image of the two buildings housing the TTP, Client, and Merchant. A 641 m optical fiber link connects the parties.

It is similar to classical digital payments, but replaces the one-time payment token $P$ by a sequence $|P\rangle$ of quantum states. That is to say, $\kappa(C, M_i, P)$ becomes $\kappa(C, M_i, |P\rangle)$ and steps 2–5 from Fig. 2 are modified as follows:

2. The TTP generates a random bitstring $b$ and a random conjugate basis-string $\mathcal{B}$ of length $\lambda$. Each bit $b_j$ is encoded onto a quantum state prepared in $\mathcal{B}_j$, where $j \in \{1; \ldots; \lambda\}$. This constitutes the classical description $(b, \mathcal{B})$ of the quantum token $|P\rangle$, which the TTP stores under the Client's ID $C_{\text{ID}}$ (e.g., let $\lambda = 4$ with the basis $\mathcal{B}_j \in \{+/-; 0/1\}$ such that $(b, \mathcal{B}) = (0101, 0011)$ would result in $|P\rangle = "|+\rangle|-\rangle|0\rangle|1\rangle"$). The length $\lambda$ depends on the tolerated success probability of an attack and the number of available merchants.

3. Upon receiving $|P\rangle$, the Client chooses the Merchant $M_i$ out of a database that was securely pre-shared with the TTP. Next, they calculate $m_i = MAC(C, M_i)$, which is the output tag of an i.t.-secure Message Authentication Code (MAC)[38–42] that takes the secret token $C$ and the chosen Merchant's public ID $M_i$ as input (see Methods). The Client interprets $m_i$ as a basis-string and privately measures the whole sequence $|P\rangle$ according to $m_i$. The resulting string of measurement outcomes $\kappa_i \xleftarrow{m_i} |P\rangle$ constitutes the cryptogram.

4. The Client sends $\kappa_i$ along with their ID $C_{\text{ID}}$ to the Merchant, who forwards this together with its $M_i$ as $\{\kappa_i, M_i, C_{\text{ID}}\}$ to the TTP for verification.

5. To authorize the purchase, the TTP looks up $C$ and $(b, \mathcal{B})$, and calculates $m_i = MAC(C, M_i)$ for the Client's ID . The TTP accepts the transaction if and only if $(\kappa_i)_j = b_j$ when $(m_i)_j = \mathcal{B}_j$. The TTP rejects otherwise.

The protocol's security depends on the upper bound of the success probability to produce two valid, distinct cryptograms $\kappa_i$ and $\kappa_j$ for two distinct Merchants $M_i$ and $M_j$; we call this $p_d$ (c.f. following two sections). Another possible attack is to forge an output tag, such that $MAC(C, M_i) = MAC(C, M_j) \Leftrightarrow m_i = m_j \Leftrightarrow \kappa_i = \kappa_j$; we call the respective probability $p_t$.

In an i.t.-secure MAC, $p_t = 1/|m| = |M|/|C| = 1/\sqrt{|C|}$, where $|m|, |M|$ and $|C|$ refer to the cardinality of the MAC, the Merchant ID and the Client's secret token respectively. Here we assume that $|m| = |M| = \sqrt{|C|}$.

Since $p_d$ and $p_t$ should be of the same order of magnitude we choose $p_d \approx p_t = 1/\sqrt{|C|}$. This will yield the number $N$ of quantum states necessary to verify one bit of the cryptogram. As the bit length of any MAC is defined as $\log_2(|m|)$, the entire length of the quantum token will be given by $\lambda = N \cdot \log_2(|m|) = N \cdot \log_2(\sqrt{|C|})$. Any additional parameter that should be committed to during the transaction (e.g., payment amount) can be added as an input to the MAC function.

Just like QKD provides i.t.-security for key exchanges such as Diffie-Hellman[43], our scheme provides i.t.-security for the one-time property of cryptograms: while the concealment of $C$ is guaranteed by the i.t.-secure MAC, the commitment to $M_i$ is ensured by the irreversible nature of quantum measurements (see Methods). Notably, our quantum commitment is not limited by the impossibility theorem of quantum bit commitment[44,45], in which one of the two parties can delay their quantum measurements in time. This is because in our protocol one of the interacting parties is assumed to be honest (the TTP).

We note that our implementation contrasts with those of QKD schemes in two ways. First, the choice of measurement basis is deterministic as opposed to random. This effectively commits the purchase to a given Client token and Merchant. Second, the measurement bases are never publicly revealed, which has the interesting benefit of hiding the Merchant that was chosen by the Client until verification is required[32].

## Loss-dependent security

Although the security of commitment is guaranteed by the laws of quantum mechanics in theory, certain considerations have to be taken into account in a practical setting.

Due to imperfections of real devices (inaccurate state preparation, lossy quantum channels, non-unit detection efficiency), some quantum states will divert from their ideal classical descriptions, or get lost along the way. In fact, some bits in step 5 will be unequal, although measured in the same basis (i.e., $(\kappa_i)_j \neq b_j$ when $(m_i)_j = \mathcal{B}_j$) and the protocol would abort even though it was followed honestly. This is why we have to allow for errors and losses during the verification procedure. In turn, a malicious party can exploit this new allowance to circumvent the commitment or double-spend the cryptogram.

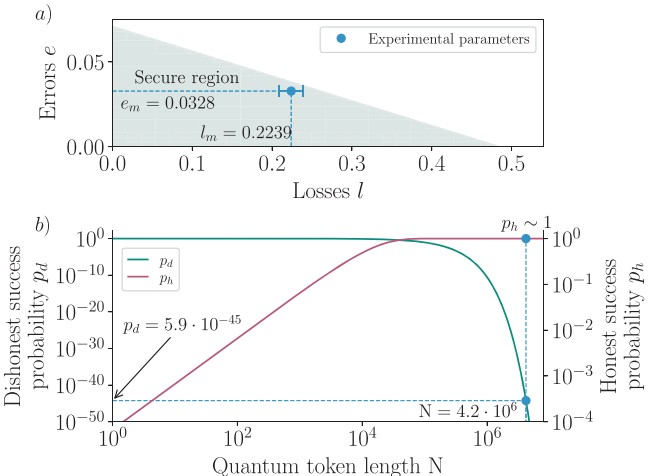

**Fig. 4 | Security for experimental quantum cryptograms. a** The semidefinite programming framework extracts a secure region of operation (turquoise) as a function of errors and losses. Our measured experimental performance ($e_m = 0.0328 \pm 0.0001$; $l_m = 0.2239 \pm 0.0150$) is indicated by the blue dot, and lies within the secure region. Error bars propagate Poisson errors on coincidence counts. **b** The dishonest success probability $p_d$ (green, upper bound) and honest success probability $p_h$ (red, lower bound) are displayed as a function of the number of quantum states $N$ required to verify one bit of the cryptogram. These are derived using a Chernoff bound argument (see Supplementary Information)[54]. As an example, an experimental token containing $\lambda = N = 4.2 \times 10^6$ quantum states (vertical blue dashed line) achieves an honest success probability very close to $p_h \sim 1$ and a dishonest success probability $p_d = 5.9 \times 10^{-45}$.

As an example, assume that the TTP tolerates as many as 50% losses. A malicious Client could then measure half of the quantum token $|P\rangle$ in the basis for $M_0$ and the other half in the basis for $M_1$, effectively creating two successfully committed tokens. While double-spending is certainly possible with a tolerated loss rate of 50%, we use semidefinite programming to identify combinations of error and loss rates for which an attack can still be detected. Intuitively, the derivation involves searching for the cheating strategy that minimizes the malicious party's introduction of excess errors and losses in the protocol (see Methods). We note that, to the best of our knowledge, such powerful loss-dependent attacks were not considered in previous quantum token implementations[24,32].

### Experimental demonstration

We implement our quantum-digital payment scheme over the deployed optical fiber link depicted in Fig. 3. The TTP employs a Spontaneous Parametric Down Conversion (SPDC) source to create a pair of polarization-entangled photons in the state $|\Psi^-\rangle = (|HV\rangle - |VH\rangle)/\sqrt{2}$. The TTP keeps one of these photons and employs a 50/50 beamsplitter to probabilistically direct it to one of two polarization projection stages, measuring its polarization in either the linear H/V ($+$) or diagonal D/A ($\times$) basis. This creates the random classical description $(b, \mathcal{B})$ and remotely imprints the payment token $|P\rangle$ onto the second photon.

The payment token is sent to the Client, located in another building, through a 641 m optical fiber link. Using a half-wave plate, the Client commits to exactly one Merchant from the set $\{M_0, M_1\}$ by measuring either in the H/V basis for $m_0 = MAC(C, M_0)$ or in the D/A basis for $m_1 = MAC(C, M_1)$. In this way, the Client retrieves the classical cryptogram $\kappa(C, M_i, |P\rangle)$, and forwards it to the Merchant, who is, for convenience, located in the same laboratory. Note that in the case of more than two merchants, the token is split into several sub-tokens that are each measured either in H/V or D/A. We discuss how to adapt the token length in the following section.

At any later time, the Merchant transmits the *cryptogram* received by the Client back to the TTP, using a classical channel that links the two buildings. The TTP finally checks the compatibility of $(b, \mathcal{B})$ with $M_i$, $C$ and $\kappa_i$, and accepts or rejects the requested transaction.

We repeatedly execute the experiment for both commitments in H/V and D/A. The average measured error rate is $1.45 \pm 0.01\%$ for H/V and $3.28 \pm 0.01\%$ for D/A. The overall losses, combining the deployed fiber link and the Client's setup (including detection efficiency), are estimated at $22.40 \pm 1.50\%$, while the multiphoton emission probability, measured through a correlation measurement, is $6.76 \pm 0.12\%$. The detail of such values are presented in the Supplementary Information.

With a maximum measured error rate of $e_m = 3.28 \pm 0.01\%$ (D/A) and losses of $l_m = 22.40 \pm 1.50\%$, we lie within the calculated secure region as depicted in Fig. 4a. In fact, according to our semidefinite programs (see Methods), a cheating party will introduce errors larger than $e = 3.79 \pm 0.22\%$ when double-spending with the same amount of claimed losses $l = l_m$. With $e_m < e$ and $l_m < l$ by two standard deviations, we therefore demonstrate that a TTP can allow for honest experimental imperfections while ensuring protection against malicious parties.

The i.t.-secure implementation of our protocol depends only on statistical fluctuations arising from the finite number of generated quantum states: a malicious party may indeed successfully cheat by introducing fewer losses or errors than the expected asymptotic values displayed in Fig. 4a.

We use the Chernoff bounds from Fig. 4b to estimate the dishonest success probability $p_d$ associated with the number $N$ of quantum states required to verify one bit of the cryptogram. We also determine the probability $p_h$ that the protocol does not abort when followed honestly, which tends to $p_h \sim 1$ as $N$ is increased.

## Discussion

We propose and demonstrate a form of quantum payment that guarantees the one-time nature of purchases with i.t.-security. By increasing the length of the quantum token, the cheating probability becomes arbitrarily low in the presence of experimental imperfections such as noise and losses. The implementation does not require any challenging technology on the Client's side, besides single-photon detection.

While typical contactless payment delays are of the order of seconds, our quantum communication and verification scheme provides i.t.-security within a few tens of minutes. These limitations are, however, only technological: quantum communication rates can be improved by using brighter quantum sources, while the verification delay originates from the correction of time-tagging drifts between the two buildings (see Methods). Indeed, brighter sources of entangled photon pairs have already been demonstrated, which could decrease the quantum token transmission time to under a second[46].

We finally note that practical digital payment schemes must allow for rejected payments without compromising the Client's sensitive data. In our scheme, the adversary can compromise the payment token $|P\rangle$ sent over the quantum channel, the cryptogram sent over the classical channel, or the Client's choice of $M_i$.

In the first two cases, quantum mechanics will ensure that the TTP recognizes the malformed cryptogram and rejects the payment with arbitrarily high probability. The transaction may then be restarted. However, an i.t.-secure MAC must not re-use the key $C$ (see Methods), which is why we propose the use of an $n$-time-secure MAC to overcome this obstacle. This allows re-using $C$ as an input for the following payments, which imposes a finite, arbitrary bound on the number of purchases[39,40]. When $C$ is consumed, a new $C$, shared between Client and TTP, must then be generated. We can amend our protocol such that the number of purchases is not bounded by the MAC function, by growing $C$ during the payment process: when the Client receives a new quantum token $|P\rangle$, we append additional quantum states for QKD, and use the cryptogram $\kappa$ for authentication. To protect against the third case, it must be ensured that the Client's choice of $M_i$ is independent of any external bias. This can for example be guaranteed if a secure database of

Merchants is initially distributed along with $C$ and the Client chooses freely without any prior communication with the Merchant. Alternatively, the Merchant may send their ID to the Client, who uses the local database as a 2nd factor authentication.

Our protocol's relaxed implementation requirements with respect to previous proposals, together with its error-tolerance, facilitate its deployment in mid-term quantum networks. Classical networks host applications beyond mere communication tasks. Similarly, a future quantum internet will necessitate the maturation of various quantum primitives and applications beyond QKD[9,47]. Our scheme advances the field of quantum payment schemes towards mid-term practical relevancy.

# Methods

## Cryptogram
A cryptogram is a cryptographic function that secures tokenized payments (e.g., online, contactless, and in-app-payments) against double-spending[15,16]. The actual cryptographic mechanism varies per payment-network, but a typical procedure is *challenge-response*. Here, the Client is not only in possession of a payment token, but also shares a secret key with the TTP[35]. During the payment, the Merchant generates a pseudo-random value (called a *nonce*), and sends it to the Client who encrypts it with this key (typically, symmetric encryption with ≥128-bit key strength is used). The resulting *cryptogram* is sent alongside the payment meta-data (e.g., merchant ID, amount, etc.) to the Merchant, who forwards it to the TTP. As the TTP is in possession of the key, they are able to decrypt and prove the correctness of the nonce for the given payment at the Merchant. Spending the token for another transaction is impossible under the assumptions of computationally secure encryption.

## i.t.-secure MAC
A *Message Authentication Code (MAC)* is a function $f(H, k, m) \mapsto y$. Based on a pseudo-randomized function $H$ – typically a hash function –, it takes a secret key $k$ and message $m$ as inputs, and outputs some authentication tag. A hash function is defined as a function that maps a set of arbitrary length to a finite set $H : \{0,1\}^* \mapsto \{0,1\}^n; n \in \mathbb{N}$. Hence, hash functions are non-injective by definition, and thus collisions, such that $f(H, k, m) = f(H, k, m'); m \neq m'$ can occur (given that $k$ remains secret). In an i.t.-secure MAC, the probability of such a collision is bound to $1/\sqrt{|k|}; k = \{0,1\}^l$, where $l \in \mathbb{N}$ is some security parameter. This is similar to the probability of finding the decryption key for a given one-time pad. Different such schemes exist, in which a key $k$ can either only be used once[37,38,41], a finite amount of times[39,40], or outputted tag length is variable[41,42].

## Semidefinite programming
Our quantum-cryptographic security proof involves optimizing over semidefinite positive objects to find an adversary's optimal cheating strategy. Semidefinite programming provides a suitable framework for this, as it allows to optimize over semidefinite positive variables, given linear constraints[48]. Most of the time, these variables are density matrices, measurement operators, or more general completely-positive trace-preserving maps[49]. Semidefinite programs present an elegant dual structure, which associates a dual maximization problem to each primal minimization problem. The optimal value of the primal problem then upper bounds the optimal value of the dual problem, allowing to prove tight bounds on the adversarial cheating probability (see ref. [26] for instance).

## Optimal cheating strategy
Using semidefinite programs, we search for the optimal completely-positive trace-preserving quantum map which minimizes the introduction of noise and losses for an adversary attempting to double-spend the cryptogram. The security analysis takes into account multiphoton emission, and assumes the absence of coherence between

photon number states. The latter is justified by the fact that SPDC produces states of the form $\sum_{n=0}^{\infty} \sqrt{c_n}|n\rangle_1|n\rangle_2$ in the $\{|n\rangle\}$ photon number basis[50], which leaves the individual subsystems in states of the form $\sum_{n=0}^{\infty} c_n|n\rangle\langle n|$. The resulting cheating strategy is fairly intuitive when considering two extreme cases: when the tolerated error rate is zero, the malicious party splits the quantum token into two equal parts, and measures each half in a different basis. This leads to two tokens that are committed to different merchants with zero error, but with 50% losses on each. On the other hand, when the tolerated losses are zero, the malicious party measures all states in a basis that is rotated by 22.5° with respect to the H/V basis. Such a measurement will identify the correct encoded bit with a probability of ~85.4%. The actual optimal cheating strategy corresponding to our experimental parameters is a combination of these two extreme strategies.

## State generation
An SPDC process in a periodically-poled KTP crystal is pumped with a continuous-wave 515 nm laser, yielding a pair of polarization-entangled and color-entangled photons. One photon is emitted at around 1500 nm, while its orthogonally polarized counterpart is emitted around 785 nm. Experimental demonstrations using a similar entanglement design were demonstrated in refs. [51,52]. Since the spectral bandwidths of the two SPDC processes are not equal, a tunable EXFO bandpass filter is inserted into the 1500 nm arm to equalize them and enhance the entanglement visibility. In order to render the two photons temporally indistinguishable, an unpoled KTP crystal of half the length of the ppKTP crystal, with axes rotated by 90° with respect to the ppKTP axes, is inserted.

## Single-photon detection
After the optical fiber link, the 1500 nm photons are detected with PhotonSpot superconducting nanowire single-photon detectors, with efficiencies around 93% (see Supplementary Information for detail), while the 785 nm photons are locally detected in the TTP's laboratory using Roithner avalanche single-photon detectors, with efficiencies around 50%. A set of paddles, inserted before the polarization measurement, are used to compensate for polarization drifts over the fiber link.

## Data post-processing
The TTP's and Client's single-photon detectors are connected to two different Time-Tagging Modules (TTM). In order to recover coincidences between the two buildings, careful synchronization of the two modules is required: first, the internal clocks of the respective modules bear an offset with respect to one another, due to the photon travel time through the optical fiber link. Second, the cycles of the internal clocks of the two TTMs drift with slightly different rates, resulting in an offset drift over time. Finally, there is an electronic delay due to different detector response times, and the TTMs only record time tags relative to the time they were activated. All these factors were corrected with our post-processing code.

## Heralded second-order correlation function measurement
To measure the heralded second-order correlation function $g_h^{(2)}(\tau)$, the 1500 nm (telecom) photons created by our SPDC source are sent directly to an InGaAs detector (idler detector labeled $D_i$), while the 785 nm photons are routed to a 50/50 fiber beamsplitter, with both outputs connected to one detector each (labeled $D_1$ and $D_2$). $g_h^{(2)}(\tau)$ can be written as[53]

$$g_h^{(2)}(\tau) = \frac{N_i \cdot N_{i12}(\tau)}{N_{i1}(0) \cdot N_{i2}(\tau)}, \tag{1}$$

where $N_i$ is the total number of events detected in the telecom detector during the measurement integration time; $N_{i1}(0)$ are the twofold

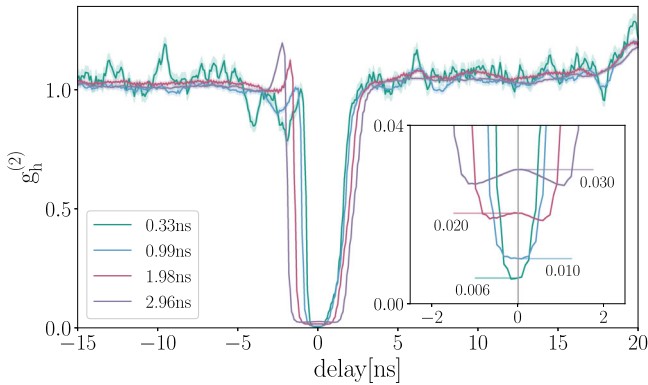

**Fig. 5 | Heralded second-order correlation function.** Data were acquired for 60 mins at a pump power of 35 mW. Coincidences were calculated using four different time windows: 0.33 ns (green), 0.99 ns (blue), 1.98 ns (red), 2.96 ns (violet). From this measurement, we determine $g_h^{(2)}(0) = 0.030\,10(14)$ for the coincidence window used in the implementation of the protocol. Shaded areas represent error-propagated uncertainties due to Poissonian photon statistics.

coincidence events between the telecom detector and D1 at 0 delay; $N_{i2}(\tau)$ are the 2-fold coincidence events between the telecom detector and D2 at delay $\tau$; and $N_{i12}(\tau)$ are the 3-fold coincidences between all 3 detectors with at delay $\tau = 0$ between the telcom detector and D1, and delay $\tau$ to D2. Pumping the SPDC source with 35 mW, data was acquired for about 60 min. $g_h^{(2)}(\tau)$, with coincidence time windows of 0.33 ns, 0.99 ns, 1.98 ns, 2.96 ns as shown in Fig. 5. A source dominated by single-photons has a $g_h^{(2)}(0) < 0.5$, with $g_h^{(2)}(0) = 0$ for a true single-photon source. From our measurements with a coincidence window of 2.96 ns, which is close to the combined jitter of the SPADs and coincidence logic and therefore the most meaningful value, we determined $g_h^{(2)}(0) = 0.030\,10(14)$.

## Data availability

The data generated in this study have been deposited in the Zenodo database under the accession code https://doi.org/10.5281/zenodo.7979319.

## Code availability

The code used in this study has been deposited in the Zenodo database under the accession code https://doi.org/10.5281/zenodo.8020667.

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

## Acknowledgements

P.S., J.K., E.S., M.-C.R., T.G., M.B., and P.W. acknowledge funding from the European Union's Horizon Europe research and innovation program under Grant Agreement No. 101114043 (QSNP) and No. 899368 (EPIQUS), along with the Quantum Flagship under grant No. 820474 (UNIQORN). The authors also acknowledge support from the Austrian Science Fund FWF through [F7113] (BeyondC), and [FG5] (Research Group 5); by the AFOSR via FA9550-21- 1-0355 (QTRUST); from the Austrian Federal Ministry for Digital and Economic Affairs, the National Foundation for Research, Technology and Development and the Christian Doppler Research Association. A.T. acknowledges support from the European Union's Horizon 2020 research and innovation program under the Marie Skłodowska-Curie grant agreement no. 801110 and the Austrian Federal Ministry of Education, Science, and Research (BMBWF). For the purpose of open access, the authors have applied a CC BY public copyright licence to any Author Accepted Manuscript version arising from this submission. P.S. thanks Teodor Strömberg, and the authors thank Fofy Setaki, for fruitful discussions.

## Author contributions

M.B. and P.W. conceived the project. J.K., E.S., T.G., and M.B. derived the security analysis. P.S., J.K., E.S., M.-C.R., A.T., and M.B. performed the experiment and analyzed the experimental data. P.S., J.K., and M.-C.R. wrote the code required to run the experiment and process the experimental data. T.G. researched and explained the relevant classical digital payment schemes. E.S., M.B., and T.G. designed the final protocol. P.S., J.K., E.S., T.G., and M.B. wrote the manuscript, with inputs from M.-C.R., A.T., and P.W.

## Competing interests

P.W., M.B., T.G., E.S., and P.S. are employees of the University of Vienna, which has applied for a patent (EP 23168897.9) for the use of a quantum payment token scheme with P.W., M.B., T.G., E.S., and P.S. listed as inventors. The remaining authors declare no competing interests.
