## [Peer Review File · Nature Communications]

Demonstration of quantum-digital paymentsREVIEWER COMMENTS

Reviewer #1 (Remarks to the Author):

The authors of the manuscript “Demonstration of quantum-digital payments” propose and demonstrate a novel quantum information-based use-case, namely digital payment as the title says. Well aware of the impossibility of Information Theoretically Secure (ITS) bit commitment (Refs [42,43] in the manuscript they propose utilizing a limited version of that primitive that involves three parties a trusted bank, a client and a merchant. They claim ITS security based on a commitment with a trusted sender and never made public “sifting” in the quantum phase, while attempting to ward off potential malicious attempts by the client and the merchant(s).

In my opinion their attempt is not successful as I try to demonstrate below. However, the application proposed is interesting and it requires either additional assumptions or better explanation as I do not exclude that I have not been able to fully understand the setting of the authors.

I have major remarks, minor remarks and some comments. All these are listed below:

Major remarks

1.The proposed cryptographic scheme utilizes the digital identities C of the client and M of the merchant. It is not stated how these are obtained by the client. As for C this can be obtained by the client upon the initial interaction with the bank, which is not problematic in principle. However, it is not clear how the digital identity M of the merchant is made available. One possibility is that some sort of a Public Key Infrastructure (PKI) is used. However, this requires trust in still another party that runs the PKI and the impossibility of any malicious actor to impersonate a merchant (pretend to be a merchant) by using an otherwise legitimate M . All this could be solved by means on an identity authentication primitive, but it is well known that such is impossible on the ITS level.

2.The security of the quantum transfer scheme is based on the comparison of the pairs (l_h, e_h) and (l_d, e_d) . I find that this is the weakest part of the proposed crypto approach. It seems to me that the authors implicitly assume that the pair (l_h, e_h) is universally known, also to the trusted bank. However a malicious client can perform the transfer from a much nearer location, corresponding to a lower l_h and thus break the security and potentially overspend. I ask the authors to clarify how they enforce that a malicious client cannot modify the pair (l_h, e_h) to his advantage. In QKD, the traditional there eavesdropper is not bound to any values of this pair.

Minor remarks

1.The scheme (if it works) is limited to a number of transactions as the number of ITS MACs that can be generated depends on the initial transaction of the client with the bank: Namely what is the length of

the random bit string (the authentication key) that the client obtains from the bank? In contrast to standard QKD this key is not regenerated. While in principle there is no problem for the client to get a significant amount of such key upon the initial transaction with the bank, keeping a large amount of secure key is a liability. It can well be copied together with C by an external malicious party and then all security will be lost. This is in contrast with QKD, where the originally distributed authentication key is replaced already after the first session by newly generated key, which renders QKD also forward secure and independent (except for a "short interval") from relying on the initial secret.

2. Inequality (3) in the supplement: the direction of this inequality needs to be reversed.

Comment

The authors consider an authentication matrix (beginning of page 4 of the supplement). While this is possible, I would point out that ITS authentication does not need to be done using matrices. Other algorithms are also well known. However, I would underline that this is non-essential for the paper and that the authors never claim that ITS authentication can only be done by using a matrix.

Reviewer #2 (Remarks to the Author):

In this work the authors propose a demonstration of quantum-digital payments.

The manuscript is well written and technically sound. After some improvements I can recommend it for publication.

- Some variables are not defined carefully in the current version, please check all definitions.

- I suggest to add a brief paragraph on the application of the results in a quantum Internet setting, see the suggested reference: Advances in the Quantum Internet, Communications of the ACM, DOI: 10.1145/3524455

Reviewer #3 (Remarks to the Author):

The manuscript demonstrates a quantum-digital payment scheme over a 641 m urban optical fibre link, and show its robustness to noise and loss-dependent attacks. Different from previous works, the scheme proposed in the manuscript can prevent banknote counterfeiting and double-spending attacks independent of long-term quantum storage or a complex network of trusted agents and authenticated

channels. It is an interesting idea to construct the scheme by applying the no-cloning property of quantum states, message authentication codes through hashing, and trusted third parties to provide the superiority mentioned above.

The paper is clearly written and I believe the results are publishable in Nature Communications, provided the following points are addressed:

1. In the manuscript the definition of security of quantum-digital payments seems not clearly demonstrated. The authors should show the definition from the perspective of cryptography so that the proof is logically complete.
2. There are six steps in the proposed quantum-digital payment scheme, and the manuscript reads 'Only one authenticated communication (between the client and their payment provider) has to take place at an arbitrary prior point in time'. Why do other steps not require authenticated channels?
3. Another question about security is whether the merchant can be malicious. If the merchant tampers with when transferring the message, the TTP will not pass the payment. Meanwhile, the client can no longer pay this token. Should this scenario be considered?
4. The manuscript reads 'the concealment of C is guaranteed by the i.t.-secure MAC', which means that the merchant has no knowledge of C. However, in step 5, the TTP needs to 'look up C'. How could TTP know about the C?
5. Another question about the protocol is what is the value of the quantum token C. Is it fixed and decided beforehand or can it be determined as the transaction begins?
6. I am very curious about the difference between quantum-digital payment and quantum digital signature which has unforgeability and nonrepudiation with information-theoretic security. From the perspective of the protocol in this paper, the work is to prevent merchants from tampering with consumption information and the emergence of users' second consumption. However, at present, it seems that the protocol does not prevent users from repudiation, that is, clients have clearly completed consumption, but they can deny to the TTP that they have generated this consumption. Could the authors add some sentences to clarify this problem?

7. The authors use the MAC m_i to authenticate the client's token C and identity of the merchant M_i . By encoding m_i to the measurement based on $\frac{1}{\sqrt{2}}(|P\rangle)$, which is similar to the encoding manipulation in quantum key distribution, m_i is actually encrypted between the client and TTP, and thus, the information of C and M_i is secret and integrated, i.e., cannot be eavesdropped or tampered with. This structure is partly analogous to but different from a recently proposed efficient QDS scheme in [Natl. Sci. Rev. 10, nwac228 (2023), <https://doi.org/10.1093/nsr/nwac228>], where a QKD-like process and MAC are also utilized to protect the secrecy and integrity of messages. Are there any deep connections between the two schemes? Could the authors compare this work and supplement some illustration in the manuscript?

8. In the last, I am a little bit confused about the security analysis in the supplementary materials which is, I think, not clear enough for the general readers. For example, what are the classical answers $\{|a_0\rangle, |a_1\rangle, |\emptyset\rangle\}$? Some of the notations do not appear in the main text or in the supplementary material before, and I think the authors need to check whether the notations for security analysis are mentioned earlier in the text.

Point-by-point reply to Reviewer comments for “Demonstration of quantum-digital payments”

We thank the Reviewers for their careful analysis of our work and their constructive feedback. We hereby provide detailed point-by-point replies to all the raised comments and suggestions.

In response to Reviewer 1:

Reviewer 1: The authors of the manuscript Demonstration of quantum-digital payments propose and demonstrate a novel quantum information-based use-case, namely digital payment as the title says. Well aware of the impossibility of Information Theoretically Secure (ITS) bit commitment (Refs [42,43] in the manuscript they propose utilizing a limited version of that primitive that involves three parties a trusted bank, a Client and a Merchant. They claim ITS security based on a commitment with a trusted sender and never made public sifting in the quantum phase, while attempting to ward off potential malicious attempts by the Client and the Merchant(s). In my opinion their attempt is not successful as I try to demonstrate below. However, the application proposed is interesting and it requires either additional assumptions or better explanation as I do not exclude that I have not been able to fully understand the setting of the authors. I have major remarks, minor remarks and some comments. All these are listed below:

Our reply: We thank the Reviewer for their very insightful assessment of our work, and for highlighting their interest in the proposed application. Alongside the minor remarks, we tackle the Reviewer’s two major remarks as follows:

- We amend the Main Text and propose an ITS solution to the first remark.
- We clarify why our approach does *not* assume any of the problematic statements mentioned in the second remark (i.e. we do not assume that the adversary is bound by a value of (l_h, e_h)). We amend the Supplementary Information accordingly.

1. Reviewer 1: The proposed cryptographic scheme utilizes the digital identities C of the Client and M of the Merchant. It is not stated how these are obtained by the Client. As for C this can be obtained by the Client upon the initial interaction with the bank, which is not problematic in principle. However, it is not clear how the digital identity M of the Merchant is made available. One possibility is that some sort of a Public Key Infrastructure (PKI) is used. However, this requires trust in still another party that runs the PKI and the impossibility of any malicious actor to impersonate a Merchant (pretend to be a Merchant) by using an otherwise legitimate M . All this could be solved by means on an identity authentication primitive, but it is well known that such is impossible on the ITS level.

Our reply: We thank the Reviewer for raising this significant point, which we have not explicitly addressed in the previous version of our paper. Clearly, the Client has to obtain the Merchant’s ID (M_i) in an IT secure way, otherwise the Merchant could maliciously send another M' to the Client, or a malicious man-in-the-middle between Client and Merchant could alter the Merchant’s ID sent to the Client and trigger a payment to the wrong account.

First, we’d like to stress that a payment token – as the one we propose in our paper – cannot protect against malicious Merchants that have a valid ID but act dishonestly by not sending the goods or providing services after the payment is performed. *Second*, however, we’d like to show how our protocol *is able* to protect the payment from maliciously acting Merchants or man-in-the-middle attacks, and *third*, why the channels used to transfer the cryptogram κ (e.g. Client \rightarrow Merchant; Merchant \rightarrow TTP) do not have to be authenticated.

The key idea to cope with these attacks is to ensure that nobody can bias the Client’s choice of M_i . There are several ways to do that, of which we imagine the simplest as a database of Merchant IDs that is stored on the Client’s device and received from the TTP during the initial secure – i.e., *authenticated* – sharing of C . So the Client can choose M_i out of this database independently, before any interaction with the (potentially dishonest) Merchant. Doing so, the Merchant *must* choose the correct M_i to send along with the cryptogram to the TTP, otherwise the payment will be rejected. Hence, there is no authentication needed on the channels: Client \rightarrow Merchant \rightarrow TTP.

We have added these clarifications to the Main Text’s “Digital Payments”, “Quantum Advantage” and “Discussion” sections, thus ensuring that the Reader is aware of such solutions. While these might seem impractical at first glance, one can imagine scenarios where this works perfectly well in the considered use case, notably without changing any assumptions on the protocol, e.g.,

- The database could be used for 2-factor-authentication. In this case, the Merchant can send their ID to the Client, who has to confirm M_i by checking name, location, etc. associated with the sent ID from the local database.

- The database can be updated whenever there is a secure channel to the TTP, e.g. when receiving a new C . We would like to stress, however, that such an update can be established at any point in time using e.g. QKD between TTP and Client.

2. Reviewer 1: The security of the quantum transfer scheme is based on the comparison of the pairs (l_h, e_h) and (l_d, e_d) . I find that this is the weakest part of the proposed crypto approach. It seems to me that the authors implicitly assume that the pair (l_h, e_h) is universally known, also to the trusted bank. However a malicious Client can perform the transfer from a much nearer location, corresponding to a lower l_h and thus break the security and potentially overspend. I ask the authors to clarify how they enforce that a malicious Client cannot modify the pair (l_h, e_h) to his advantage. In QKD, the traditional there eavesdropper is not bound to any values of this pair.

Our reply: We thank the Reviewer for giving us the opportunity to clarify our security analysis: namely, explaining why our comparison-based approach remains *strictly secure* in the presence of dishonest parties. We believe that part of the misunderstanding was triggered by a confusing use of the notations (l_h, e_h) and (l_d, e_d) , which we have now removed and unified as (l, e) .

As a first step of any implementation, the secure region of operation (i.e. the set containing all pairs (l, e) for which a malicious behavior can be detected) is calculated by all parties locally and individually using a semidefinite program (SDP). This program computes the optimal cheating strategy (i.e. the strategy which minimizes the introduction of noise and losses) of an all-powerful adversary having access to lossless and noiseless quantum channels, and yields the border line of pairs $\{(l, e)\}$ separating the secure and insecure regions shown in Fig 4.a. Any pair situated above the line allows for successful attacks that cannot be detected (insecure region), while all pairs below the line cannot have led to a successful attack (secure region). Crucially, since this derivation does not involve any experimental measurement or calibration of the infrastructure’s noise and losses, it cannot be tampered with (as each party’s local computer is trusted), and we can thus securely assume that the secure/insecure regions of operation are publicly known by all parties.

As a second step, the protocol takes place as described in the manuscript. Upon verification, the TTP checks that the experimentally measured pair (which we now label (l_m, e_m) instead of (l_h, e_h) to not mislead the Reader into thinking that these values are honest), lies within the secure region of operation. We emphasize once again that, as long as (l_m, e_m) lies within this region, no attack, even exploiting perfect channels or closer locations, could have succeeded in forging the cryptogram. In the case where (l_m, e_m) lies outside of this region, then the protocol aborts. This implies that there are no additional security

assumptions with respect to QKD: the dishonest party is not bound to any value of noise and losses, and the protocol will abort if the measured parameters lie outside of the publicly-known secure region.

We have added these clarifications to Section III.A of the Supplementary Information, and changed all notations to (l, e) and (l_m, e_m) accordingly.

3. Reviewer 1: The scheme (if it works) is limited to a number of transactions as the number of ITS MACs that can be generated depends on the initial transaction of the Client with the bank: Namely what is the length of the random bit sting (the authentication key) that the Client obtains from the bank? In contrast to standard QKD this key is not regenerated. While in principle there is no problem for the Client to get a significant amount of such key upon the initial transaction with the bank, keeping a large amount of secure key is a liability. It can well be copied together with C by an external malicious party and then all security will be lost. This is in contrast with QKD, where the originally distributed authentication key is replaced already after the first session by newly generated key, which renders QKD also forward secure and independent (except for a short interval) from relying on the initial secret.

Our reply: We thank the Reviewer for the careful assessment of our protocol and pointing out this possible security flaw. In order to avoid any misunderstandings, we would first like to emphasize that there is only one authentication key, namely C , which is privately linked to the public Client ID at the TTP. Furthermore we would like to acknowledge that the necessity for long time storage can indeed be avoided by using QKD before every run of our protocol.

For our envisioned use-case, however, we believe that it is important to ensure IT security in a practical/flexible environment, rather than ensuring absolute security in every aspect (e.g. secure storage devices). While the absolute security of the Client’s device could be achieved by using many authenticated channels when a payment is performed, we favor the practical advantage of not requiring any authenticated (quantum or classical) channel after the initial distribution of C (and the list of Merchant IDs M_i).

However, we would like to argue that C can be refreshed at any time if needed, by establishing a secure connection to the TTP, e.g. via QKD, i.e. it is possible to redo the initial step as often as necessary/preferred. We would also like to point out that this additional QKD step could also take place when the Client receives a new quantum token $|P\rangle$, by receiving more quantum states (e.g., $|P\rangle |C_{i+1}\rangle$) and authenticating with the TTP along with the payment token κ . Although κ is sent over an insecure classical channel, C is i.t.-securely grown by the knowledge of C_i , that cannot be altered. We described this scenario in the paragraph “Discussion”.

Finally, today’s payment infrastructure generally relies on trust assumptions with respect to hardware devices (e.g. chip cards, encrypted storage on phones). Therefore, relying on a secure storage of C after the IT-secure sharing seems to be a fair assumption.

4. Reviewer 1: Inequality (3) in the supplement: the direction of this inequality needs to be reversed.

Our reply: We thank the Reviewer for thoroughly checking the Supplementary Information. We would like to point out that, in Supplementary Section I.A, we have taken the standard convention of John Watrous’s lecture notes that the primal problem is a maximization problem and the associated dual problem is a minimization problem. In that scenario, we believe that the inequality is correct as it is (i.e. $s_p \leq s_d$), since the set of all dual values (optimal and non-optimal) should upper bound the set of all primal values (optimal and non-optimal). We do agree with the reviewer however, that the paragraph leading up to this equation contained an inconsistency, which we now fixed to read:

“The Lagrange multiplier method allows to find the local extremum of a constrained function. The optimal value s_p of the primal problem therefore ~~lower upper~~ bounds the optimal value s_d of the dual problem, while the optimal value of the dual ~~upper lower~~ bounds that of the primal. This property is known as weak duality, and may be simply expressed as:”

However, we have added a note in Section III.A to make the Reader aware that, for the sake of intuition in our security proof, we formulate our primal problem as a minimization problem (and hence the dual as a maximization problem).

5. Reviewer 1: The authors consider an authentication matrix (beginning of page 4 of the supplement). While this is possible, I would point out that ITS authentication does not need to be done using matrices. Other algorithms are also well known. However, I would underline that this is non-essential for the paper and that the authors never claim that ITS authentication can only be done by using a matrix.

Our reply: We thank the Reviewer for pointing out that the authentication matrix used in our Supplementary Information is only an example, and not necessary for ITS authentication. We opted for this particular description to simplify the derivation of the cheating probability. We amended the Supplementary Information accordingly, to stress that such a matrix is not necessary.

In response to Reviewer 2:

Reviewer 2: In this work the authors propose a demonstration of quantum-digital payments. The manuscript is well written and technically sound. After some improvements I can recommend it for publication.

Our reply: We thank the Reviewer for their positive evaluation of our work and for recommending its publication in *Nature Communications*. We answer all the raised comments in a point-by-point manner below.

- 1. Reviewer 2:** Some variables are not defined carefully in the current version, please check all definitions.

Our reply: We have indeed found inconsistencies in the notations. We have carefully checked both Main Text and Supplementary Information, and added clarifications when relevant.

- 2. Reviewer 2:** I suggest to add a brief paragraph on the application of the results in a quantum Internet setting, see the suggested reference: Advances in the Quantum Internet, Communications of the ACM, DOI: 10.1145/3524455

Our reply: We thank the Reviewer for their suggestion to provide the Reader with a broader picture of the protocol's role in a future quantum internet. The following added paragraph in the "Discussion" section now addresses this point:

"Our protocol's relaxed implementation requirements with respect to previous proposals, together with its error-tolerance, facilitate its deployment in mid-term quantum networks. Classical networks host applications beyond mere communication tasks. Similarly, a future quantum internet will necessitate the maturation of various quantum primitives and applications beyond QKD [Gyongyosi2022, Wehner2018]. Our protocol advances the field of quantum payment schemes towards mid-term practical relevancy."

In response to Reviewer 3:

Reviewer 3: The manuscript demonstrates a quantum-digital payment scheme over a 641 m urban optical fibre link, and show its robustness to noise and loss-dependent attacks. Different from previous works, the scheme proposed in the manuscript can prevent banknote counterfeiting and double-spending attacks independent of long-term quantum storage or a complex network of trusted agents and authenticated channels. It is an interesting idea to construct the scheme by applying the no-cloning property of quantum states, message authentication codes through hashing, and trusted third parties to provide the superiority mentioned above.

The paper is clearly written and I believe the results are publishable in *Nature Communications*, provided the following points are addressed:

Our reply: We thank the Reviewer for their careful assessment of our work, and for supporting its publication in *Nature Communications*. We provide point-by-point replies to all comments below.

- 1. Reviewer 3:** In the manuscript the definition of security of quantum-digital payments seems not clearly demonstrated. The authors should show the definition from the perspective of cryptography so that the proof is logically complete.

Our reply: We thank the Reviewer for raising this important point. In order to not interrupt the flow of the first sections of the Main Text, we had originally postponed the rigorous security definition to the end of the “Results” section, to Figure 4.b, and to the Supplementary Information. However, we agree that the security parameter should be defined right after the quantum protocol is defined, and have therefore shifted this discussion to the earlier “Quantum advantage” section:

“The protocol’s security is defined as the upper bound p_d on the success probability of an attack consisting in producing two cryptograms κ_0 and κ_1 for two distinct Merchants M_0 and M_1 that both pass the TTP’s verification test, as well as the probability p_t of forging the output tag of the MAC. In an i.t.-secure MAC, $p_t = 1/\sqrt{|C|}$ [41]. Since p_d and p_t should be of the same order of magnitude, we choose $p_d = |M|/|C|$ and a MAC with output length $|M| = \sqrt{|C|}$. The total length of the quantum token is finally given by $\lambda = N \cdot \sqrt{|C|}$. Any additional parameter that should be committed to during the transaction (e.g., payment amount) can be added as an input to the MAC

function.”

- 2. Reviewer 3:** There are six steps in the proposed quantum-digital payment scheme, and the manuscript reads Only one authenticated communication (between the Client and their payment provider) has to take place at an arbitrary prior point in time. Why do other steps not require authenticated channels?

Our reply: The initial distribution of C is indeed the only step of that requires a classical authenticated channel. All other channels do not have to be authenticated, as any malicious behavior can be detected thanks to our security analysis, in which case the protocol can abort. In order to provide intuition about this, we have added a section to the Supplementary Information that reads the following:

Compromising classical channels

We have two classical channels in our protocol, namely CH2 (Client \rightarrow Merchant) and CH3 (Merchant \rightarrow TTP) in FIG.1 of the main text. Since both of them are untrusted, it is possible for a malicious third party to intercept them and modify the cryptogram $\kappa(C, M_i, |P\rangle)$ towards another merchant M'_i on CH2 or change the merchant's Id M_i towards another merchant's M'_i on CH3.

CH2 *To be accepted by the TTP, the attacker has to find another $\kappa'(C, M'_i, |P\rangle)$ for the Client's secret C that commits the purchase to another Merchant M'_i . This is impossible for two reasons:*

1.) the attacker would need to determine C from the Client to calculate a second measurement bases $m' = \text{MAC}(C, M'_i)$, which is supposed to be securely distributed between Client and TTP. It is impossible to determine C from κ as the function of the measurement basis $\text{MAC}(C, M_i)$ is information-securely irreversible – and its output is additionally hidden in the quantum measurement and therefore unknown to the attacker.

2.) even if the attacker would have access to C for some reason – e.g. by accessing the Client's memory –, he would require the classical description of quantum token $|P\rangle$, since it is already measured and quantum measurements are destructive. However, the classical description is only known by the TTP and never communicated.

CH3 *To change M_i that is communicated together with κ , the attacker has to find M'_i that generates the same measurement bases $m' = \text{MAC}(C, M'_i)$ that was used to generate κ . To*

do so, he would need access C , which is supposed to be securely distributed between Client and TTP. However, even if he would have access, the chances of finding a collusion such that $MAC(C, M_i) = MAC(C, M'_i)$ for a given C are exponentially low due to the information-theoretic nature of the MAC function.

If the Merchant requires instant notification of the payments acceptance, however, this channel requires authentication s.t. the Client could not alter this message.

Please note, that both attacks can be performed by a malicious Merchant as well – who has access to both channels and is supposed to be untrusted – but fail for the same reason.

Compromising the quantum channel

A significant advantage of our scheme is that it is preferable but not necessary to authenticate the quantum channel used to distribute $|P\rangle$. Let us suppose that a malicious party intercepts the quantum states $|P\rangle$ and sends their own quantum states $|P'\rangle$ to the Client instead. After the Client measures $|P'\rangle$ in the basis $MAC(C, M_i)$, they will hold the cryptogram $\kappa' = \kappa(C, M_i, |P'\rangle)$. If κ' reaches the TTP, the transaction will be declined since $\kappa(C, M_i, |P'\rangle) \neq \kappa(C, M_i, |P\rangle)$ (within the error/loss tolerance allowed by the security analysis). This means, that the Client as well as the TTP will be able to detect that the quantum states have been tampered with and that precautions should be taken.

Another possible cheating strategy is for the malicious party to use the quantum token $|P\rangle$ themselves and measure it in another basis than the Client had intended. However, the malicious party does not know C , since it was securely distributed only between Client and TTP, and is thus unable to determine any measurement basis m_j that will be accepted for the Merchant that they choose.

Compromising all channels simultaneously

Let us now suppose that a malicious party intercepts $|P\rangle$ on the quantum channel, replaces it with another quantum token $|P'\rangle$, and waits for the honest Client to send the resulting κ' on the classical channel. If the Client would measure $|P'\rangle$ in a basis that is dependent on C in a simple way, e.g. $m_i = M_i \oplus C$ then the malicious party gains knowledge of C , and can then substitute the Client's identity in multiple transactions. This is why we use a MAC instead: even if the malicious third party gets hold of κ' and, by knowing $|P'\rangle$, deduces the measurement basis m_i , they are unable to retrieve C , because of the information theoretically secure nature of the used MAC, i.e. because the number of collision ensures that no cheating strategy would

be better than guessing. Thus again, the TTP (and subsequently the Client) realise that something is wrong, while the secret Client token C remains hidden. Depending on the nature of the MAC the token C may resist a certain amount of failures, before it has to be exchanged.”

3. Reviewer 3: Another question about security is whether the Merchant can be malicious. If the Merchant tampers κ when transferring the message, the TTP will not pass the payment. Meanwhile, the Client can no longer pay this token. Should this scenario be considered?

Our reply: In the case of a malicious Merchant (or any attacker on the classical channels; see previous comment), any attempt on changing κ will result in rejection of the payment by the TTP. Notably, any payment token κ can only be used once to prevent double-spending of the Client. Hence, in order to proceed with the payment, the Client has to restart the protocol, i.e. generate the measurement basis $m = \text{MAC}(C, M_i)$ receive and measure a new quantum token $|P'\rangle$ to retrieve κ' . This is perfectly valid, as the reason of rejection could be due to technical issues on any of the channels.

However, this yields a potential “Denial of Service” attack: an attacker or malicious Merchant repetitively alters κ and the Client consumes all of his secret token C . In this case, we assume the Client to abort the payment after a given number of trials.

We consider such an attack outside the scope of our protocol, as such attacks cannot be circumvented by cryptographic means.

4. Reviewer 3: The manuscript reads the concealment of C is guaranteed by the i.t.-secure MAC, which means that the Merchant has no knowledge of C . However, in step 5, the TTP needs to look up C . How could TTP know about the C ?

Our reply: C is the secret key that is pre-shared between the TTP and the Client before the transaction starts. It is held private by both the TTP and the Client. Thus if the TTP sees a transaction that is performed under the public ID C_{ID} of the specific Client, they can look up C on their own private databank. However, C is indeed kept hidden from all other parties involved, which is achieved using the i.t.-secure MAC function. We have now clarified this in the protocol description of the Main Text, by stating that the Client ID is sent along with κ when a transaction is requested. Note that this public ID of the Client is equivalent to e.g. the card number of a credit card etc. as long as it is unique.

5. Reviewer 3: Another question about the protocol is what is the value of the quantum token C . Is it fixed and decided beforehand or can it be determined as the transaction begins?

Our reply: The secret identification token C is established at the initialisation of the protocol, thus it is a fixed classical bitstring distributed via QKD or another i.t.-secure distribution method. It cannot be altered afterwards, however, the Client will use different parts for every transaction, as a certain part of C may only be used once or a limited amount of times, depending on the cryptographic function used. The quantum token $|P\rangle$, on the other hand, is randomly generated upon the start of every payment procedure. The classical description of $|P\rangle$ is afterwards securely stored at the TTP. Note that the lengths of both C and $|P\rangle$ are estimated from the security parameters in the manner given in our answer to Comment #1.

Noteworthy – given C is still sufficient in size – C can be grown during the payment process. When the Client receives a new quantum token $|P\rangle$, by receiving more quantum states (e.g., $|P\rangle|C_{i+1}\rangle$) and authenticating with the TTP along with the payment token κ . Although κ is sent over an insecure classical channel, C is i.t.-securely grown by the knowledge of C_i , that cannot be altered. We described this scenario in the paragraph “Discussion”.

6. Reviewer 3: I am very curious about the difference between quantum-digital payment and quantum digital signature which has unforgeability and non-repudiation with information-theoretic security. From the perspective of the protocol in this paper, the work is to prevent Merchants from tampering with consumption information and the emergence of users’ second consumption. However, at present, it seems that the protocol does not prevent users from repudiation, that is, Clients have clearly completed consumption, but they can deny to the TTP that they have generated this consumption. Could the authors add some sentences to clarify this problem?

Our reply: We thank the Reviewer for pointing out this interesting similarity between quantum payment tokens and quantum signature schemes (QDS). We believe that it is indeed possible to map one approach to the other, although there exist several non-trivial mappings of three parties (TTP, Client, Merchant) onto the typical QDS parties (Alice, Bob, Charlie) (present in Ref 1 and Ref 2 for instance), that provide unforgeability and non-repudiation for a payment.

We consider the following to be the most intuitive: Let us assume that the message to be signed is the transaction of the Client’s money from their bank account to the Merchant’s account, e.g. `sign(K_A , "send 100 EUR to Bob")`, where K_A is Alice’s secret key. Intuitively, the Client takes the role of Alice, while the Merchant slips into Bob’s role, and the TTP becomes Charlie, who is trusted by

both Alice and Bob. Now, *unforgeability* and *non-repudiation* map as follows:

unforgeability is when the Merchant (Bob) – or in our case any attacker on the classical link Client \rightarrow Merchant \rightarrow TTP – modifies the cryptogram κ in an attempt to change the payment’s details, e.g. transfer the money to a malicious bank account (and additionally later deny having received the money). Our protocol can prevent any changes of the transaction, as described earlier in the answer to Comment #2 and in the Supplementary’s section V.

non-repudiation is when the Client (Alice) tries to deny having spent the money to stop (e.g. to receive goods without payment) or revoke the actual bank transfer of the money in an attempt to double spend “physical” money at two Merchants. As the TTP is trusted, this renders impossible due to the shared secret C and the randomness introduced by measuring the uncloneable quantum states, of which only the TTP knows the classical description.

Of course, if the TTP is untrusted (or Charlie is untrusted in QDS), then non-repudiation doesn’t hold, since the TTP could create a payment token κ in the Client’s name.

We would like to stress that, even though it is possible to secure digital payments with QDS, our approach adds two main advantages: *First*, in contrast to QDS, we require only the classical channel TTP \leftrightarrow Client (i.e. Alice \leftrightarrow Charlie) to be authenticated for the pre-sharing of C . This greatly eases the trust assumptions. *Second*, we require only two parties to share a quantum channel (i.e. for QKD and sending $|P\rangle$ between TTP and Client), while QDS schemes typically require quantum channels and classical authenticated channels between all three parties.

7. Reviewer 3: The authors use the MAC m_i to authenticate the Clients token C and identity of the Merchant M_i . By encoding m_i to the measurement based on $|P\rangle$, which is similar to the encoding manipulation in quantum key distribution, m_i is actually encrypted between the Client and TTP, and thus, the information of C and M_i is secret and integrated, i.e., cannot be eavesdropped or tampered with. This structure is partly analogous to but different from a recently proposed efficient QDS scheme in [Natl. Sci. Rev. 10, nwac228 (2023), <https://doi.org/10.1093/nsr/nwac228>], where a QKD-like process and MAC are also utilized to protect the secrecy and integrity of messages. Are there any deep connections between the two schemes? Could the authors compare this work and supplement some illustration in the manuscript?

Our reply: We thank the Reviewer for pointing out the existence of this interesting work. For the mapping/connection between this QDS scheme and our protocol, please refer to the previous comment.

We have condensed this as a small transition between the “Digital payments” and “Quantum advantage” section:

Quantum advantage. Considering these attacks only, previous quantum digital signature schemes can provide i.t.-security [Ref1 , Ref2]. However, they typically require QKD channels and classical authentication between all three parties.

In this work, we propose a quantum solution that requires only one QKD for the initial step between Client and TTP (Step 1 in Fig. 2). It is similar to classical digital payments, but replaces the one-time payment token P by a sequence $|P\rangle$ of quantum states.[...]

Additionally, we cited the usage of the OTUH-function introduced in Ref2 as a potential MAC function for our scheme in the Methods section.

8. Reviewer 3: In the last, I am a little bit confused about the security analysis in the supplementary materials which is, I think, not clear enough for the general readers. For example, what are the classical answers $|a_0\rangle, |a_1\rangle, |\emptyset\rangle$? Some of the notations do not appear in the main text or in the supplementary material before, and I think the authors need to check whether the notations for security analysis are mentioned earlier in the text.

Our reply: We thank the Reviewer for thoroughly checking the Supplementary Information and pointing out the missing clarifications. The vectors $|a_0\rangle, |a_1\rangle$ and $|\emptyset\rangle$ form a 3-dimensional orthonormal basis, in which each basis vector denotes one possible classical outcome: a dishonest party may declare a “0”, a “1”, or report a no-detection flag denoted by \emptyset . Each outcome must be orthogonal to the other, since they are purely classical states. We have clarified this in the Supplementary Information. Additionally, we have carefully checked the Main Text and Supplementary Information for missing definitions and added them where necessary.

REVIEWERS' COMMENTS

Reviewer #1 (Remarks to the Author):

I recommend this paper for publication, possibly after a minor review. (A detailed review is given in the attachment.)

The authors of the manuscript “Demonstration of quantum-digital payments” have done a very good job in their replies to the reviewer comments and modifications to the manuscript.

My two major comments are satisfactorily answered in the Reply of the authors. I do agree with both of them. Specifically:

As long as the TTP possess the true digital identity of the Merchant and is providing this to the Client e.g. in the form of trusted database of currently valid merchants, whereby this database can be periodically updated through some form of secure channel, the concern I expressed is not a source of a real danger. What is important is that there is no reliance on some form of a public PKI for ITS security as the access to the latter can be manipulated with sufficient resources. I also agree that nothing can be done against malicious Merchants, who have a valid ID but do not supply goods. Naturally there should be mechanisms in place for Client complaints and TTP guaranteeing the transaction, even if the malicious merchant would suddenly withdraw his funds and “disappear”. A version of this type of attacks might be some malicious “Merchant” with a valid ID that hijacks the electronic presence of an honest Merchant but replaces the ID of the latter with his own. Possibly a partial protection against such scenarios is inclusion of the name of the Merchant in the protocol, not just his ID. If there is a mismatch between the ID and the Name TTP should reject the cryptogram. If the name is not of the Merchant advertised, the Client should notice and do not carry out the transaction. But OK, these are things outside the core protocol.

Concerning my second major comment, I agree with the present detailed explanation and slight change of notation: the situation has become clear to me. I must confess that I have been confused by the text and notation during the original review.

With this, I find the paper perfectly suitable for publication.

Concerning authentication, I still have some technical comments of minor overall importance but I think that these can be addressed by the authors in the final preparation of the manuscript for publication.

First I absolutely agree that “there is only one authentication key, namely C, which is privately linked to the public Client ID at the TTP”, as the authors write. I must again confess that originally I was automatically thinking of an ITS one time authentication, while the authors were addressing an N time one. Such methods do exist and that is sufficient. My remark is that, in fact the authors, quote some quite specific implementation-related approaches (Refs. [11], [12] from the supplement) and specifically Ref. [42] (from the main manuscript), which I do not understand why it is at all quoted. In fact standard methods for n-times ITS secure authentication are well-known, and, I think, best presented in the classical Reference [9] (numbering as in the supplement) – see specifically Section 4 (Authenticating multiple messages) of that publication. (Moreover these standard methods do not contradict in any way the approach of the authors.)

I also completely agree with the authors on their discussion of whether long-term keeping of key is relevant or not. This has practical, implementation consequences, but not protocol validity ones.

Several more comments in this relation. I fail to understand the text from the main text (Quantum advantage):

The protocol's security is defined as the upper bound p_d on the success probability of an attack consisting in producing two cryptograms κ_0 and κ_1 for two distinct Merchants M_0 and M_1 that both pass the TTP's verification test, as well as the probability p_t of forging the output tag of the MAC. In an i.t.-secure MAC, $p_t = 1/\sqrt{|C|}$ [41]. Since p_d and p_t should be of the same order of magnitude, we choose $p_d = |M|/|C|$ and a MAC with output length $|M| = \sqrt{|C|}$. The total length of the quantum token is finally given by $\lambda = N \cdot \sqrt{|C|}$. Any additional parameter that should be committed to during the transaction (e.g., payment amount) can be added as an input to the MAC function.

I must admit that here that I fail to understand the exact meaning of this text on several levels. First the paragraph from the classic reference [41] that I believe the authors refer, reads as

To create an authentication system which is unbreakable with certainty p , we can simply choose T to have at least $1/p$ elements, and let F be a strongly universal₂ class of hash functions from M to T . If we let H' be the subset of H which maps m to $f(m)$, we see that the only information that the forgers have available is that the secret function is one of the functions in H' . However, the definition of strongly universal₂ implies that for any m' distinct from m , the proportion of functions in H' which map m' to any particular tag t' is $1/|T|$. Since $|T| \geq 1/p$, any choice the forger makes has no more than a probability of p of being correct.

At least this text is very clear in contrast to the somewhat not obviously clear definitions of p_d and p_t . What is exactly the meaning the probability of forging a tag? Perhaps finding a valid tag for a given message? Second. I can agree that the key space is equal to the number of all 2-universal hashing functions in this case (each function has to be indexed by a key). But what about the "output length" $|M|$? First of all it cannot be the length of the string. All statements are about the number of elements of a set, and the notation $| \cdot |$ is used for that purpose. But for me, more curiously $|M|$ does not come about in Ref [41] and I fail to see how it is derived here. I am curious how the authors come to this conclusion. It would be good if they give some explanation and correspondingly change the text. (A final technical comment in this respect: The symbol N - number of quantum signals - is defined much later in the text and it remains unclear to the reader at this point what it meaning is. I guess this is an artifact of moving around text in the new version.)

Finally there are some typesetting glitches in the related text in the supplement:

Some of the fonts for T and K in the equation appear to be wrong:

To generate an authentication tag for a message $m \in \mathcal{M}$ and a given key $k \in \mathcal{K}$, one takes the corresponding cell $t \in \mathcal{T}$ as an output. Assuming a uniform distribution of \mathcal{K} and generating a new \mathcal{T} for every message $m \in \mathcal{M}$, the probability of forging a valid authentication tag is $p_t = 1/|\mathcal{T}|$ if the key is only used once.

The tag space $|\mathcal{T}|$ depends on the message- and key space. For a message space of size $|\mathcal{K}| = |\mathcal{M}|^2$, the probability of forging a valid authentication tag is

$$p_t = \frac{1}{|\mathcal{T}|} = \frac{|\mathcal{M}|}{|\mathcal{K}|} = \frac{1}{\sqrt{|\mathcal{K}|}}$$

Reviewer #3 (Remarks to the Author):

I have another small suggestion, whether it needs to be modified depends on the authors. The model of quantum digital payment proposed by the authors is based on the concept of quantum token, i.e., quantum money. This model is more similar to banknotes and credit cards, that is, the money is saved in the client. The client selects merchants and then consumes the currency, and TTP verifies the currency. However, there is another kind of digital payment, similar to electronic bank transfer and Amazon online shopping. Only the client initiates a consumption request (including the amount and merchant information) to TTP, and TTP directly transfers the money from the client's account to the merchant's account. In this way, currency will not pass through customers, and the attack of double-spending naturally does not exist. Therefore, in my opinion, the author's model is part of digital payment (similar to digital currency and credit card), and does not include the type of electronic transfer and online shopping. I think is it better to change the quantum digital-payment in the title to quantum currency or quantum credit card? This small issue does not affect my positive attitude towards the publication of this article.

Point-by-point reply to Reviewer comments for “Demonstration of quantum-digital payments”

We thank the Reviewers for the thorough re-evaluation of our work and for their constructive feedback. We hereby provide detailed point-by-point replies to the raised comments and suggestions.

In response to Reviewer 1:

Reviewer 1: The authors of the manuscript “Demonstration of quantum-digital payments” have done a very good job in their replies to the Reviewer comments and modifications to the manuscript. My two major comments are satisfactorily answered in the Reply of the authors. I do agree with both of them. Specifically: As long as the TTP possess the true digital identity of the Merchant and is providing this to the Client e.g. in the form of trusted database of currently valid merchants, whereby this database can be periodically updated through some form of secure channel, the concern I expressed is not a source of a real danger. What is important is that there is no reliance on some form of a public PKI for ITS security as the access to the latter can be manipulated with sufficient resources. I also agree that nothing can be done against malicious Merchants, who have a valid ID but do not supply goods. Naturally there should be mechanisms in place for Client complaints and TTP guaranteeing the transaction, even if the malicious merchant would suddenly withdraw his funds and “disappear”. A version of this type of attacks might be some malicious “Merchant” with a valid ID that hijacks the electronic presence of an honest Merchant but replaces the ID of the latter with his own. Possibly a partial protection against such scenarios is inclusion of the name of the Merchant in the protocol, not just his ID. If there is a mismatch between the ID and the Name TTP should reject the cryptogram. If the name is not of the Merchant advertised, the Client should notice and do not carry out the transaction. But OK, these are things outside the core protocol.

Our reply: We are pleased to read that the implemented changes have cleared out the previous misunderstandings. We agree that including the name of the Merchant along with his ID in the protocol would partially protect against the mentioned hijacking attack. To clarify this point, we had amended the protocol accordingly in the last version.

Reviewer 1: Concerning my second major comment, I agree with the present detailed explanation and slight change of notation: the situation has become clear to me. I must confess that I have been confused by the text and notation during the original review. With this, I find the paper perfectly suitable for publication.

Our reply: We thank the Reviewer again for pointing out this inconsistency in our original version, thus significantly clarifying the security analysis for the Reader.

Reviewer 1: Concerning authentication, I still have some technical comments of minor overall importance but I think that these can be addressed by the authors in the final preparation of the manuscript for publication. First I absolutely agree that “there is only one authentication key, namely C, which is privately linked to the public Client ID at the TTP”, as the authors write. I must again confess that originally I was automatically thinking of an ITS one time authentication, while the authors were addressing an N time one. Such methods do exist and that is sufficient. My remark is that, in fact the authors, quote some quite specific implementation-related approaches (Refs. [11], [12] from the supplement) and specifically Ref. [42] (from the main manuscript), which I do not understand why it is at all quoted. In fact standard methods for n-times ITS secure authentication are well-known, and, I think, best presented in the classical Reference [9] (numbering as in the supplement) – see specifically Section 4 (Authenticating multiple messages) of that publication. (Moreover these standard methods do not contradict in any way the approach of the authors.)

I also completely agree with the authors on their discussion of whether long-term keeping of key is relevant or not. This has practical, implementation consequences, but not protocol validity ones.

Our reply: In light of this remark, we have included the suggested standard reference ([9] from the Supplementary Information) to the Main Text.

Reviewer 1: Several more comments in this relation. I fail to understand the text from the main text (Quantum advantage):

The protocol's security is defined as the upper bound p_d on the success probability of an attack consisting in producing two cryptograms κ_0 and κ_1 for two distinct Merchants M_0 and M_1 that both pass the TTP's verification test, as well as the probability p_t of forging the output tag of the MAC. In an i.t.-secure MAC, $p_t = 1/\sqrt{|C|}$ [41]. Since p_d and p_t should be of the same order of magnitude, we choose $p_d = |M|/|C|$ and a MAC with output length $|M| = \sqrt{|C|}$. The total length of the quantum token is finally given by $\lambda = N \cdot \sqrt{|C|}$. Any additional parameter that should be committed to during the transaction (e.g., payment amount) can be added as an input to the MAC function.

I must admit that here that I fail to understand the exact meaning of this text on several levels. First the paragraph from the classic reference [41] that I believe the authors refer, reads as

To create an authentication system which is unbreakable with certainty p , we can simply choose T to have at least $1/p$ elements, and let F be a strongly universal₂ class of hash functions from M to T . If we let H' be the subset of H which maps m to $f(m)$, we see that the only information that the forgers have available is that the secret function is one of the functions in H' . However, the definition of strongly universal₂ implies that for any m' distinct from m , the proportion of functions in H' which map m' to any particular tag t' is $1/|T|$. Since $|T| \geq 1/p$, any choice the forger makes has no more than a probability of p of being correct.

At least this text is very clear in contrast to the somewhat not obviously clear definitions of p_d and p_t . What is exactly the meaning the probability of forging a tag? Perhaps finding a valid tag for a given message? Second. I can agree that the key space is equal to the number of all 2- universal hashing functions in this case (each function has to be indexed by a key). But what about the "output length" $|M|$? First of all it cannot be the length of the string. All statements are about the number of elements of a set, and the notation $| \cdot |$ is used for that purpose. But for me, more curiously $|M|$ does not come about in Ref [41] and I fail to see how it is derived here. I am curious how the authors come to this conclusion. It would be good if they give some explanation and correspondingly change the text.

Our reply: We thank the Reviewer for both comments. Regarding the first, indeed there has been a mix-up of notations: By $|C|$ or $|M|$, it is customary to denote the size of the key- and message-space respectively and not the bit-length of the corresponding element. We have therefore changed the mathematical definitions accordingly. Regarding the second point, we hope that the following text clarifies the definitions of p_d and p_t :

The protocol's security depends on the upper bound of the success probability to produce two valid, distinct cryptograms κ_i and κ_j for two distinct Merchants M_i and M_j ; we call this p_d (c.f. following two sections). Another possible attack is to forge an output tag, such that $\text{MAC}(C, M_i) = \text{MAC}(C, M_j) \Leftrightarrow m_i = m_j \Leftrightarrow \kappa_i = \kappa_j$; we call the respective probability p_t . ~~The protocol's security is defined as the upper bound p_d on the success probability of an attack consisting in producing two cryptograms κ_0 and κ_1 for two distinct Merchants M_0 and M_1 that both pass the TTP's verification test, as well as the probability p_t of forging an output tag of the MAC.~~

In an i.t.-secure MAC, $p_t = 1/|m| = |M|/|C| = 1/\sqrt{|C|}$, where $|m|$, $|M|$ and $|C|$ refer to the cardinality of the MAC, the Merchant ID and the Client's secret token respectively. Here we assume that $|m| = |M| = \sqrt{|C|}$. Since p_d and p_t should be of the same order of magnitude we choose $p_d \approx p_t = 1/\sqrt{|C|}$. This will yield the number N of quantum states necessary to verify one bit of the cryptogram. As the bit length of any MAC is defined as $\log_2(|m|)$, the entire length of the quantum token will be given by $\lambda = N \cdot \log_2(|m|) = N \cdot \log_2(\sqrt{|C|})$. ~~In an information-theoretic secure (i.t.-secure) MAC, $p_t = 1/\sqrt{|C|}$ [41]. Since p_d and p_t should be of the same order of magnitude, we choose $p_d = |M|/|C|$ and a Message Authentication Code (MAC) with output length $|M| = \sqrt{|C|}$. The total length of the quantum token is finally given by $\lambda = N \cdot \sqrt{|C|}$.~~

Reviewer 1: A final technical comment in this respect: The symbol N - number of quantum signals - is defined much later in the text and it remains unclear to the reader at this point what it meaning is. I guess this is an artifact of moving around text in the new version.

Our reply: We have solved this issue by adding an explanatory sentence earlier in the text (see quoted text from previous comment).

Reviewer 1: Finally there are some typesetting glitches in the related text in the supplement: Some of the fonts for T and K in the equation appear to be wrong:

To generate an authentication tag for a message $m \in \mathcal{M}$ and a given key $k \in \mathcal{K}$, one takes the corresponding cell $t \in \mathcal{T}$ as an output. Assuming a uniform distribution of \mathcal{K} and generating a new \mathcal{T} for every message $m \in \mathcal{M}$, the probability of forging a valid authentication tag is $p_t = 1/|\mathcal{T}|$ if the key is only used once.

The tag space $|\mathcal{T}|$ depends on the message- and key space. For a message space of size $|\mathcal{K}| = |\mathcal{M}|^2$, the probability of forging a valid authentication tag is

$$p_t = \frac{1}{|T|} = \frac{|\mathcal{M}|}{|\mathcal{K}|} = \frac{1}{\sqrt{|K|}}$$

Our reply: We thank the Reviewer for noticing this typesetting inconsistency and have corrected the issue in the equation, which now reads

$$p_t = \frac{1}{|\mathcal{T}|} = \frac{|\mathcal{M}|}{|\mathcal{K}|} = \frac{1}{\sqrt{|\mathcal{K}|}}$$

In response to Reviewer 3:

Reviewer 3: I have another small suggestion, whether it needs to be modified depends on the authors. The model of quantum digital payment proposed by the authors is based on the concept of quantum token, i.e., quantum money. This model is more similar to banknotes and credit cards, that is, the money is saved in the client. The client selects merchants and then consumes the currency, and TTP verifies the currency. However, there is another kind of digital payment, similar to electronic bank transfer and Amazon online shopping. Only the client initiates a consumption request (including the amount and merchant information) to TTP, and TTP directly transfers the money from the client's account to the merchant's account. In this way, currency will not pass through customers, and the attack of double-spending naturally does not exist. Therefore, in my opinion, the author's model is part of digital payment (similar to digital currency and credit card), and does not include the type of electronic transfer and online shopping. I think is it better to change the quantum digital-payment in the title to quantum currency or quantum credit card? This small issue does not affect my positive attitude towards the publication of this article.

Our reply: We thank the Reviewer for the overall positive assessment of our work and pointing out this interesting remark. While it is true, that in our precise implementation of the protocol we do not explicitly take into account such online payments as Amazon online shopping or electronic bank transfer, we do believe that our scheme could be implemented in such scenarios as well. Even if the Client makes their request directly to the TTP, bypassing the Merchant, there has to be some means of identification. This is especially true, if we assume that the communication channel between Client and TTP is potentially compromised (after the initial sharing of C).

Regarding the suggested terms, we believe that

- the term *currency* is incorrect, since the protocol actually implements a secure currency transfer scheme rather than a currency itself.
- the term *credit card* narrows the possible applications of our protocol as it is too specific.

We therefore think that the current title adequately describes our protocol.